# Interpreting and Generating Gestures with Embodied Human Computer Interactions

James Pustejovsky
Brandeis University
Waltham, MA
jamesp@brandeis.edu

Nikhil Krishnaswamy
Colorado State University
Fort Collins, CO
nkrishna@colostate.edu

Ross Beveridge
Colorado State University
Fort Collins, CO
ross.beveridge@colostate.edu

Francisco R. Ortega
Colorado State University
Fort Collins, CO
fortega@colostate.edu

Dhruva Patil
Colorado State University
Fort Collins, CO
patil.dhruva@gmail.com

Heting Wang
University of Florida
Gainsville, FL
heting.wang@ufl.edu

David G. McNeely-White
Colorado State University
Fort Collins, CO
david.white@colostate.edu

## ABSTRACT

In this paper, we discuss the role that gesture plays for an embodied intelligent virtual agent (IVA) in the context of multimodal task-oriented dialogues with a human. We have developed a simulation platform, VoxWorld, for modeling and building *Embodied Human-Computer Interactions (EHCI)*, where communication is facilitated through language, gesture, action, facial expressions, and gaze tracking. We believe that EHCI is a fruitful approach for studying and enabling robust interaction and communication between humans and intelligent agents and robots. Gesture, language, and action are generated and interpreted by an IVA in a *situated meaning context*, which facilitates grounded and contextualized interpretations of communicative expressions in a dialogue. The framework enables multiple methods for performing evaluation of gesture generation and recognition. We discuss four separate scenarios involving the generation of non-verbal behavior in dialogue: (1) deixis (pointing) gestures, generated to request information regarding an object, a location, or a direction when performing a specific action; (2) iconic action gestures, generated to clarify how (what manner of action) to perform a specific task; (3) affordance-denoting gestures, generated to describe how the IVA can interact with an object, even when it does not know what it is or what it might be used for; and (4) direct situated actions, where the IVA responds to a command or request by acting in the environment directly.

## CCS CONCEPTS

• **Human-centered computing; HCI**;

## KEYWORDS

gesture interpretation, gesture generation, multimodal embodiment, simulation, virtual agent, situated grounding

## 1 INTRODUCTION

Human-to-human communication is essential to daily communication. Getting to this level of communication with intelligent avatars interacting with humans requires further work than the current approach to conversational agents (CAs), such as Apple's Siri or Amazon's Alexa or approaches to embodied conversational agents (ECAs). We present a direction forward when dealing with the challenges confronting the generation and recognition of non-verbal behavior in the context of multimodal interactions involving an Intelligent Virtual Agent (IVA) with a human. This research shows that a bidirectional IVA is required for a realistic interaction. The simulation platform for modeling such interactions is called *Embodied Human-Computer Interactions (EHCI)*.

The system used is called VoxWorld, which is a multimodal dialogue system enabling communication through language, gesture, action, facial expressions, and gaze tracking, in the context of task-oriented interactions. A multimodal simulation is an embodied 3D virtual realization of both the situational environment and the co-situated agents, as well as the most salient content denoted by communicative acts in a discourse. It is built on the modeling language VoxML [33], which encodes objects with rich semantic typing and action affordances, and actions themselves as multimodal programs, enabling contextually salient inferences and decisions in the environment. VoxWorld enables an embodied HCI by situating both human and computational agents within the same virtual simulation environment, where they share perceptual and epistemic common ground.

Within an embodied HCI, actions, gesture, language, and facial expressions are all interpreted and generated by an IVA in an environment where meaning is situationally grounded and contextualized to the discourse and updates in the environment.

This IVA is unique in that is a *symmetric model* of non-verbal behavior for the IVA. This entails being able to both recognize and generate an expression in the context of an interaction with a human partner (interlocutor). This bidirectionality to the interaction is enabled by the IVA being contextualized in an embodied interaction, where both the output to the gesture classifier and the input to the gesture generation reference the same underlying semantic representation. This is illustrated in 1, where on the left, a human is action gesturing to move an object to the left, while on the right, the IVA is performing the identical gesture.

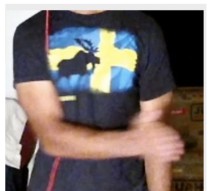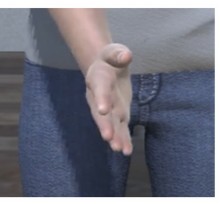

**Figure 1: Bidirectional gesture recognition and generation.**

To illustrate the role of EHCI in planning multimodal interactions, we discuss four different scenarios involving the generation of non-verbal behavior:

(1) deixis (pointing) gestures, generated to request information regarding an object, a location, or a direction when performing a specific action;

(2) iconic action gestures, generated to request clarification on how (what manner of action) to perform a specific task;

(3) affordance-denoting gestures, generated to describe how the IVA can interact with an object, even when it does not know what it is or what it might be used for;

(4) direct situated actions, where the IVA responds to a command or request by acting in the environment directly.

## 2 EMBODIED HCI

There has been a growing interest in the Human-Robot Interaction (HRI) community on how to contextually resolve ambiguities that may arise from communication in situated dialogues, from earlier discussions on how HRI dialogues should be designed [14, 20, 24, 35], modeling deixis and gaze [31], affective states in conversation [10], how perception and grounding can be integrated into language understanding [22, 28], to pedagogy [25], and recent work on task-oriented dialogues [40]. This is the problem of identifying and modifying the *common ground* between speakers [3, 8, 39, 41]. While it has long been recognized that an utterance's meaning is subject to contextualized interpretation, this is also the case with gestures in task-oriented dialogues.

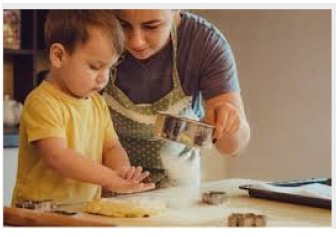

**Figure 2: Mother and child baking.**

Recently, we have argued that natural human-computer interactions involving intelligent virtual agents (IVAs) require not only that the agent itself be embodied, but that the entire *interaction* between the human and the IVA must be embodied, in order to fully establish the common ground that both agents share to communicate fluently [34] . This is referred to as an *embodied Human-Computer Interaction*, and we adopt this view for this paper.

For example, in typical task-oriented interactions between humans, (as shown in Fig. 2), actions, gesture, and language are situated within a common ground. In such situations, the common ground includes the following characteristics:

- *Co-situatedness* and *co-perception* of the agents, such that they can interpret the same situation from their respective frames of reference.

- *Co-attention* of shared situated references, allowing richer expressiveness in referring to the environment (i.e., through language, gesture, visual presentation, etc.). The human and avatar might be able to refer to objects on the table in multiple modalities with a common model of differences in perspective-relative references

- *Co-intent* or agreement of the common goals in a dialogue. It is important to recognize the intent of other agents, to facilitate the interpretation of their expressions.

In order to achieve these goals, human-computer/robot interactions requires robust recognition and generation of expressions through multiple modalities (language, gesture, vision, action); and the encoding of SITUATED MEANING: this entails three aspects of common ground interpretation: (a) the situated *grounding* of expressions in context; (b) an interpretation of the expression contextualized to the *dynamics* of the discourse; and (c) an appreciation of the *actions and consequences* associated with objects in the environment.

With this in mind, many HCI researchers have adopted the notion of "embodiment" in order to better understand user expectations when interacting with computational agents [13, 15, 26]. Embodied agents or avatars add new dimensions to human/agent interactions compared to voice- or text-only conversational agents. Embodied agents can express emotions and perform gestures, two crucial non-verbal modes of human communication. Potentially, this enables such agents to have more human-like, peer-to-peer interactions with users. Unfortunately, embodiment alone does not avoid some of the key limitations of conversational agents. Even embedded in an avatar, most agents won't know what you are pointing at. As with verbal conversations, visual communication mechanisms like gestures, expressions, and body language need to be a two-way communication.

Following [17, 27], we adopt VoxWorld as our environment supporting embodied HCI. This platform enables embodied virtual agents, who are aware not only of their own virtual space but of the physical space of the human with whom they are interacting and communicating. One such avatar, Diana, can speak, gesture, track, move, and emote [17, 27]. Diana has video and depth sensors that let her sense the physical world around her, including the user. Diana observes the user, and knows when they are attending to her. She can observe the user's emotions, and most importantly she can understand the user's gestures. As a result, visual communication joins verbal communication as a two-way process.

At the center of VoxWorld is the language VoxML [33] and the associated software, VoxSim [18]. VoxML (Visual Object Concept Modeling Language) is a modeling language for constructing 3D visualizations of concepts denoted by natural language expressions, and is used in the VoxWorld platform for creating multimodal semantic simulations in the context of human-computer and human-robot communication. VoxSim is the software that interprets the encodings of objects and events as written in VoxML, and handles visual event simulation in 3D, written with the Unity game engine.

## 3 VERBAL AND NON-VERBAL BEHAVIOR

The VoxWorld system enables multimodal communication between a human and an IVA, for task-oriented dialogue and interaction. Both human and IVA can use language, gesture, and facial expressions to communicate with each other, and actions to move the task forward; e.g., building a structure, moving objects, etc.

The human acts as a "signaler," indicating objects and actions to Diana by means of speech and gesture. The user's language (speech) is captured by Google ASR and motions are captured using a Microsoft Kinect v2 RGBD sensor. Gestures are detected in real time using custom gesture recognition software [29] and sent to the avatar. The avatar's actions, gestures, and facial expressions are displayed on the monitor for the human to see.

In the context of an embodied HCI, we consider a communicative act, $C_a$, performed by an agent, $a$, to be a tuple of expressions from the diverse modalities available to an agent, involved in conveying information to another agent. For our present discussion, let us restrict this to the modalities of a linguistic utterance, $S$ (speech), gesture, $G$, facial expression, $F$, gaze, $Z$, and an explicit action, $A$: $C_a = \langle S, G, F, Z, A \rangle$. In order to align these modalities in the state space within the dialogue manager, we assume that the common ground structure associated with a state in a dialogue or discourse, can be modeled as a state monad [4, 42]: $\mathbf{M}\alpha = State \rightarrow (\alpha \times State)$. This corresponds to those computations that read and modify a particular dialogue state. $\mathbf{M}$ is a type constructor that constructs a function type taking a state as input and returns a pair of a value and a new or modified state as output.

To illustrate the manner in which information from diverse modalities is encoded in the dialogue state, consider a communicative act that exploits a combination of speech and gesture, $(S, G)$. We can identify three configurations for how a language-gesture *ensemble* can be interpreted, depending on which modality carries the majority of semantic content: (a) language with *co-speech gesture*, where language conveys the bulk of the propositional content and gesture adds situated grounding, affect, effect, and presuppositional force [5, 23, 36, 37]; (b) *co-gestural speech*, where gesture plays this role [32]; and (c) a truly mixed modal expression, where both language and gesture contribute equally to the meaning.

In practice, while many of the interaction in our dialogue experiments have this property, the discourse narrative is broadly guided by gesture. For this reason, we will view such multimodal interactions as gesture with *co-gestural speech*. This is in fact, a subclass of *content-bearing gestures*, where gesture is used to convey the semantics normally carried by linguistic expressions. In the discussion below, we focus on the interaction of gesture, facial expressions, and gaze, with varying degrees of language.

## 3.1 Gesture

The language of gestures that the Diana IVA can recognize and interpret grew out of a year long elicitation study. 60 subjects, working in pairs, solved problems involving the construction of structures made out of blocks. The studies placed each person in a separate room with one designated as the signaler and the other the builder. Both stood in front of a table and signaler and builder were able to communicate with audio, video, or both depending upon the scenario. The three scenarios were: speech only (the builder was not allowed to see the video of the signaler); gesture only (no audio in either direction); or both and speech and gesture (subjects could both see and hear each other). In these experiments, the signaler was shown a plan; a structure that the buider needed to construct. The builder had a set of blocks. The heart of this experiment was recording how the signaler and builder communicated with each other in the course of successfully building the desired structure.

An interesting finding of this study is that subject pairs successfully completed their task using speech only, gesture only, and speech plus gesture. While the pairs completed the task about 20% faster using both modalities, pairs successfully built the desired structures in almost all cases using just gesture or just speech.

Key to the development of our gesture language was the careful review of the roughly 12.5 hours of video for repeated use of what could be considered a common gestural language. The result of the hand labeling of video initially was 24,503 distinct video segments representing what was judged to be a communicative act.

Further analyses over this large set of 24,503 segments led to us identifying 35 hand gestures, 6 arm motions and 6 body movements being used by more than 4 subjects. This labeled data became the basis for the Diana IVA gesture recognition system. Arm and body motions are captured using Kinect Skeleton data and interpreted using a hand built classifier[45]. The hand gestures are captured from Kinect depth images using a series of Resnet-style deep convolutional neural networks (DCNN)[16]. As a result, to support the non-verbal gestural communication the real-time output from these classifiers is streamed to the IVA through a blackboard.

Of the nearly 50 distinct gestures, some were predicatble and some less so. For example, using a thumbs-up for postitive acknowledgement was seen and this is not surprising. Perhaps more surprising, a majority of people used the whole body action of either stepping closer to, or away from, their table as a way of signally either a desire for engagement or completion of a task.

Note from the above that our model for how an agent should use gestures is fundamentally rooted in how people use gestures. This leads to a related requirement: for an IVA in a symmetric peer-to-peer interaction, the avatar should be capable of generating gestures at the same level that it recognizes. When the range of recognized gestures is known, this is a straightforward matter of animating those same gestures on the avatar's skeleton. Fig. 6 shows the avatar generating some of the same gestures that it can recognize.

The avatar can also generate some gesture that the human never makes: for instance, when the only manipulable objects exist in the avatar's virtual world, the human cannot reach for one of those objects. However, since the avatar can recognize when the human is pointing to one of the virtual objects, when the human does so, the avatar will reach for that object. This serves as a non-verbal "speech act" acknowledging receipt of the human's pointing gesture, and demonstrating an interpretation by generating a gesture.

For a more complex problem, such as generating a novel gesture learned in the course of an interaction, we can mirror the recognition process by breaking down the gesture generation into *hand pose generation*, where the avatar's hand tracks to a predetermined or calculated hand-pose, usually constructed relative to an object (cf. Fig 5), and an arm motion, calculated by the inverse kinematics (IK) within Unity, which causes all the arm joints between wrist and shoulder to be placed and rotated appropriately to get the hand into the required position.

## 3.2 Facial Expression and Gaze

Diana seeks to engage the user by providing non-verbal cues in the form of facial expressions. Diana has the following expressions: smile, frown, sad, frustrated, neural, and most importantly, the ability to show concentration to the user. This latter expression was

developed by surveying multiple users asking them from a set of images which one looks more "concentrated". This is relevant since Diana is performing a task and at times, looking concentrated is required. Diana had three settings: showing no emotion, mirroring the user's emotion, or displaying emotion dependent on content (the most useful scenario). The user expressions are determined using the Affectiva API.

Diana's response is determined by the user's expression and the data collected in a human-to-human builder and signaler task of 40 instances where users where asked to be either a builder (i.e., someone in charge to build a shape with the blocks) or a signaler (i.e., someone telling the builder what to build). This provides insight into the right responses that Diana must have in order to be effective, as Diana is the builder and the user is the signaler. For example, if a user is showing frustration or anger, builders showed empathy towards the user by having a gentle smile.

Diana also sees and accommodates the direction of the user's gaze. For instance, if the user looks off screen toward the left, Diana will look in that direction as well, attending to the interruption in conversation. In these circumstances, Diana will ignore speech input, acting as if she believes that anything the human says while looking in this direction is not directed toward her.

### 3.3 Action

Within VoxWorld, the primary focus has been on the generation of actions over objects performed in the simulation environment inhabited by the IVA (Diana). These include the following action primitives: *grasp*, *hold*, *touch move*, *push*, *pull*, *turn*, and *slide*. In addition, composite (or complex) actions are generated by combining these actions, using the composition mechanisms of VoxML: *put*, *place*, and *stack*. Recognition of these same actions by the IVA is in principle possible, but the focus thus far has been on recognition of multimodal communicative expressions in the dialogue.

## 4 GENERATING NON-VERBAL BEHAVIOR

### 4.1 Deictic and Action Gesture Generation

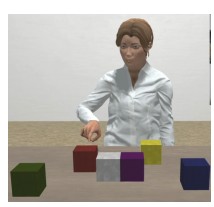

Generating non-verbal behavior in an interaction is crucial for the agent's behavior to be believable [9]. Diana can perform deictic gestures to clarify that a particular object or location is the one intended by the user (cf. Fig. 3).

Similarly, gesture can be used to direct complete actions, by identifying objects through deixis, indicating actions to be performed, and designating the intended goas location for the action.

**Figure 3: Gesture clarifies the target of an action.**

(cf. Fig. 4). Diana can generate not only individual gestures, but composite gestures, to carry out entire actions over objects, such as that illustrated below.

| Single Modality (Gesture) Imperative |
| --- |
| DIANA₁: $\mathcal{G}$ = [*points to the purple block*]$_{t1}$ 
 DIANA₂: $\mathcal{G}$ = [*makes move gesture*]$_{t2}$ 
 DIANA₃: $\mathcal{G}$ = [*points to the blue block*]$_{t3}$ |

This is rendered in VoxWorld as the gesture sequence shown in Fig. 7, which can only be interpreted relative to the situated grounding available to the IVA and human user (cf. Fig. 4).

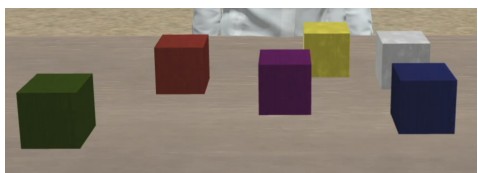

**Figure 4: Configuration of blocks on table.**

### 4.2 Affordance Gesture Generation

Objects can be analogized to each other in terms of their behaviors, and these analogies can be made more specific and accurate by comparing both the behaviors an ob-

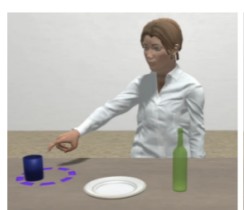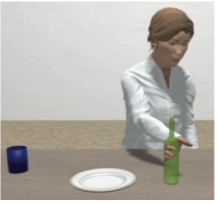

**Figure 5: Generating an affordance-denoting gesture to describe what the IVA knows about an object**

ject facilitates by virtue of its structure or purpose (*afforded behaviors*) and the spatial situations, or *habitats* in which they occur. That is, if an agent encounters an object for which it knows no name but can determine that it has a number of affordances in common with another object, it can use that second object as a starting point to reason about the first.

For example, if the agent comes across an unfamiliar object that appears to share the $H_{[2]}$ = [UP = $align(Y, \mathcal{E}_Y)$, TOP = $top(+Y)$] (upward alignment) habitat of [[CUP]][1], she can infer that it might be grasped in a similar way. Fig. 5 shows this process enacted through dialogue. In frame 1 (on the left), the human points to a new object (recognizable as a bottle, but Diana has no label associated with it). Diana reaches toward the object to acknowledge the object human's deixis. In frame 2 (on the right), Diana is demonstrating her the method of grasping she infers from that object's observed similarity to a cup.

### 4.3 Action Generation

This is quite straightforward, as it involves carrying out an action in the virtual environment, in response to the current state of the dialogue or a request from the user. When prompted to move a block, Diana responds by simply carrying out the action directly, what we call "communication by direct action".

### 4.4 Facial Expression Generation

Existing ECAs used Ekman's [11] seven basic emotions, such as [6, 7, 12, 21, 30, 38]. However, in a bidirectional IVA designed for collaoarative task building (signaler/builder), this proves a challenge.

---

[1]This can be approximately glossed as *the cup's Y-axis is aligned upward with the Y-axis of the embedding space, and if something is put inside the cup, the cup contains that thing*

**Table 1: Diana's action unit code combinations compared with other code combinations.**

| Affective States | FACS [11] | SmartBody [1] | Diana's Action Units |
|---|---|---|---|
| Joy | 6 + 12 (Happiness) | same | BrowsUp + NoseScrunch + MouthNarrow + Smile |
| Sympathy | 1 + 4 + 15 (Sadness) | 1 + 4 + 6 | BrowsOuterLower + BrowsDown + Frown + NoseScrunch + MouthNarrow |
| Confusion | 4 + 7 + 15 + 17 + 23 [2] | - | BrowsIn + Squint + NoseScrunch + JawDown |
| Concentration | - | - | BrowsUp + EyesWide |

In such environments where the IVA and the user work together towards a common goal, if Diana expressed anger when the user is showing anger as well, it will create conflict between the avatar and the user. Therefore, the performance may be hindered by this negative action of the avatar. While this is possible in a human-to-human collaboration task, in the dataset, the builder was always empathetic to the user.

Using previous work and the data analysis of CSU's EGGNOG videos [43], four responsive affective states were integrated on Diana's face. Considering the difficulty of studies in the HCI field to model empathy comprehensively [44], Diana's facial expressions used key concepts in her affect perception and generation modules were Thinking from Others Perspectives and the Appraisal Theory, components that resided in the highest level of the hierarchical model of empathy for embodied agents [44].

Diana's facial expression were design by combining knowledge gained by Ekman's units and SmartBody, along with the action units that associated with high recognition accuracy and judgment of human-likeness by [6]. Joy and sympathy were developed by combining similar definitions in the Facial Action Coding System [11]. For confusion, selected action units that were found to contribute to the perception of confusion were used. As for concentration, we proposed our creations by observing human behavior in EGGNOG [43] and asking participants in a survey to select an image that depicted confusion. Those missing action units in the character were replaced by movements of similar facial morph targets. Finally, synthesized facial expression was generated by linear movements towards pre-defined thresholds of the values of morph targets. Table 1 shows Diana's action code combinations for expressions compared with the code combinations in standard Facial Action Coding System [11] and SmartBody [1] (a characther animation application).

## 5 EVALUATION OF NON-VERBAL BEHAVIOR

Referring expressions and definite descriptions of objects in space exploit information about both object characteristics and locations. Linguistic referencing strategies can rely on increasingly high-level abstractions to distinguish an object in a given location from similar ones elsewhere, yet the description of the intended location may still be unnatural or difficult to interpret. [19] measured how humans evaluate multimodal referring expression generated by a virtual avatar. The study generated 1500 visualizations of an avatar referring to one of 6 non-distinct objects in a virtual environment. In these visualizations a target object was first indicated by a pink circle, and then the avatar referred to it using a stochastically-determined strategy. The video was then shown to annotators who rated the referring strategy shown in terms of naturalness. Referencing strategies included some combination of deictic gesture and language, from gesture only to language only to "ensemble" [32] or multimodal referring expressions consisting of a pointing gesture with an accompanying linguistic utterance. Linguistic referring strategies may indicate objects by color or location relative to other objects, and demonstratives ("this"/"that") when accompanied by a deictic gesture. By analyzing this data we were able to determine that humans consider multimodal referring expressions more natural than purely linguistic or purely gestural strategies. More descriptive language is also preferred, even in the context of a multimodal referring expression.

There are some shortcomings in this data and analysis.
(1) The data is on the small side, depending on the number of parameters that are useful for training a particular model.
(2) The data was gathered over single instances of object references in isolation. In an actual interaction (as in between people), people may use temporal or state history (e.g., "pick up the cup next to the block you just put down").
(3) The existing data describes how people interpret referring expression, but the data has not been used to train a model for generating referring expressions (generation in the original study was done stochastically). As such, data has only been gathered on interpretation, and not the other half of the problem, generation.

A sophisticated generation model requires more data than currently exists, and data that encompasses more types of information in referring strategies, entailing the need to tackle problems (1) and (2) and we are developing a study using the Diana IVA to elicit further data on how humans use multimodal referring expressions.

## 6 CONCLUSION

In this paper, we present an embodied Human-Computer Interaction framework within which language, gesture, and other non-verbal behaviors are used for communication between humans and intelligent agents. Here we have focused on generation of gesture, facial expressions, and actions, in the course of task-oriented dialogue. One unique feature of the system is the bidirectional nature of the capabilities: anything recognized is also generable by the IVA. We believe the system to be a useful platform for experimentation and evaluation studies.

## 7 ACKNOWLEDGMENTS

This work was supported by Contracts W911NF-15-C-0238 and W911NF-15-1-0459 with the US Defense Advanced Research Projects Agency (DARPA) and the Army Research Office (ARO). Approved for Public Release, Distribution Unlimited. The views expressed herein are ours and do not reflect the official policy or position of the Department of Defense or the U.S. Government. We would like to thank Ken Lai and Bruce Draper for their comments and suggestions.

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

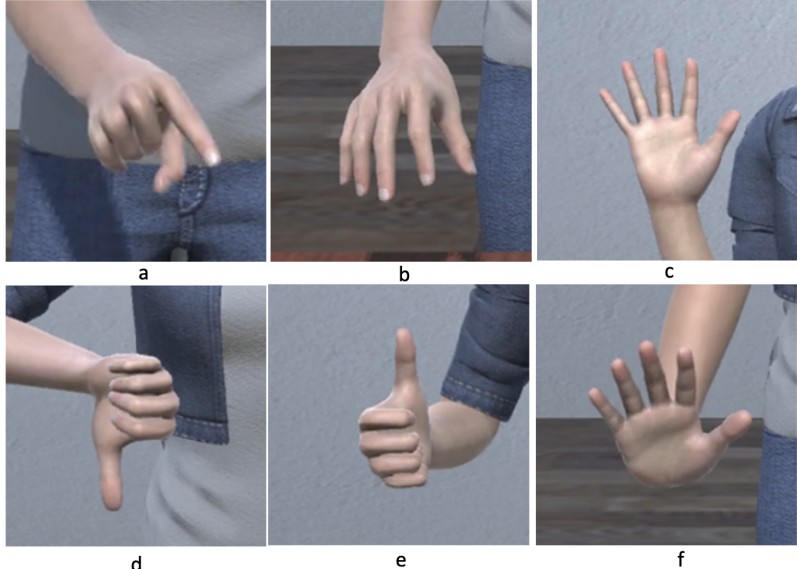

**Figure 6:  Some of the gestures generated by VoxWorld: pointing, grab, five, no, yes, push back.**

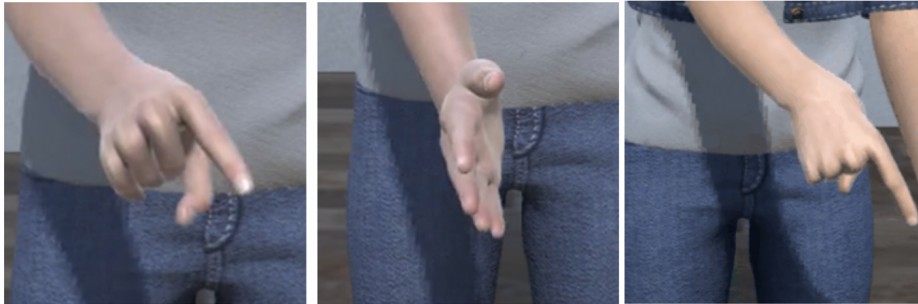

**Figure 7: Gesture generation for performing complex action.**