# OpenReview forum: "Interpreting and Generating Gestures with Embodied Human Computer Interactions"
_ACM.org/IVA/2020/Workshop/GENEA — GENEA Workshop 2020_

### Official Review · AnonReviewer4 · 2020-09-23
**The authors present an overview of complex studies. As such it remains at a very high level of descriptions. Many capabilities implementing in the virtual agent are not described.**

**Rating:** 5

**Review:**

The authors present an overview of complex studies. As such it remains at a very high level of descriptions. Many capabilities implementing in the virtual agent are not described.
The authors do not compare their work with existing ones. There is no state of the art section in the paper. Generating multimodal behaviors have been addressed a lot in the IVA community. It would be interesting to know how the proposed work differs from them, what does it bring compared to previous works… For example, regarding deixis, how does the presented work differ from works by James Lester or by Dirk Heylen? It is not clear to me why the authors do not use BML to represent the different gestures.
The authors do not mention ‘timing’ while presenting the generated behaviors for the agent, neither for the facial expressions, nor for the gesture. However, timing is crucial. A lot of work has been done in synchronizing gesture, facial expression, gaze and speech for an agent. Nothing is said in the paper how gesture, facial expression, gaze and language are synchronized with each other.
It is not clear how the agent decides on the action (behavior) to undertake? How the decision model works? It is not clear to me how ambiguity in referencing objects is dealt with.
The authors use the term ‘avatar’ or IVA or agent interchangeably. However, ‘avatar’ is used differently in the IVA community. The authors should mention at the start of the paper the use of these different terms to mean virtual agent.
In the formalization of the communicative act, how do the authors encode context? How does their formalization differ from Isabella Poggi’s work?
The authors mention three configurations to link gesture and language: how are they represented? How the generation model computes which one is to be displayed?
Regarding the third configuration, I do not understand why the truly mixed model expression requires both modalities to contribute ‘equally’ to the meaning.
There are few typos.
-	Or both and speech and gesture
-	Predicatble
-	Collaboarative
-	Charactehr

---

### Official Review · AnonReviewer1 · 2020-09-30
**The argument and approach is reasonable and viable, but it has been adopted already by lots of previous work.**

**Rating:** 4

**Review:**

The paper presents an overall approach to simulate embodied communication with am avatar in VR. The authors correctly argue for the missing ecological validity of many communication modeling attempts and that co-situatedness, co-attention, and co-intent are needed. They argue that this can be reinstantiated in a VR-based situated interaction with virtual avatars. This argument and approach is reasonable and viable, but it has been adopted already by lots of previous work (as old as 25 years). The second half of the paper presents a rather shallow way how different kinds of communicative behavior are tracked or generated, respectively. Finally, a first study is reported in which human raters judged the naturalness of the generated behavior, showing only that multimodal referring expressions are perceived as more natural. Overall, the research direction is fine and I’m glad to see people taking it further. At the same time, however, it is not yet clear what the novel contribution of the present work is going to be. Also, most of the descriptions of how behavior is generated is too superficial. Mostly the design of the behavior is described, not how it is (or can be) generated. I would recommend the authors, given the preliminary state of the work, to focus on situating the work in the larger research field and in relation to related work. The first part of the paper does a decent job in arguing for the overall approach. The latter part, however, misses a lot of related work and aptly position the work and point out its potential contributions. For example, you may want to check out:
- Leßmann, N., Kopp, S., & Wachsmuth, I. (2006). Situated interaction with a virtual human - perception, action, and cognition. In G. Rickheit & I. Wachsmuth (Eds.), Situated Communication (pp. 287-323). Berlin: Mouton de Gruyter. doi:10.1515/9783110197747.287
- Wachsmuth I, Lenzen M, Knoblich G, eds. Embodied communication in humans and machines. Oxford: Oxford Univ. Press; 2008.
- T Pfeiffer (2010). Understanding multimodal deixis with gaze and gesture in conversational interfaces.
- Callaway, J. (2001). Cosmo: A Life-like Animated Pedagogical Agent with Deictic Believability.

---

### Official Review · AnonReviewer2 · 2020-09-30
**Interpreting and Generating Gestures with Embodied Human Computer Interactions**

**Rating:** 6

**Review:**

This paper spans several different interesting issues. However it would have been better if it covered one in detail. It makes a case for Embodied HCI as a two-way situated interaction, which the paper argues for convincingly but I also believe it is widely accepted in this community. It demonstrates EHCI in a system, VOXWorld, discusses the design of the system, the formative studies/evaluations that informed that design, as well as a preliminary evaluation of the system's nonverbal behavior. Many of the issues it raises about EHCI have been addressed by others going quite far back, for example  two examples that come to mind is Hannes Vilhjalmsson's phd and the Max system by Wachsmuth, Kopp etc. The technical details and experimental are sparse which makes any systematic evaluation of technical contributions difficult.

But frankly none of the above would have mattered to this reviewer if they had succeeded in another way, because an entirely different way to look at this paper, which is how the authors at times describe their contribution, is as providing a platform for research in EHCI with virtual agents. Describing the platform, the feasibility of others using it would have made the paper a sufficient contribution to the workshop. But I am not getting a strong sense from reading the paper that it is feasible for other to use it, to incorporate their own components into it, etc. For example it is not clear that it uses SAIBA standards.

---

### Decision · Program_Chairs · 2020-10-02

**Decision:**

Accept

**Comment:**

Dear Authors,

We would like to inform you that your paper for the GENEA Workshop 2020 has been Accepted! The paper has some limitations as outlined in the reviews, but an availability of a simulation platform has a particular contribution to the IVA community.

Please try to revise your paper according to reviews and upload the camera-ready version to OpenReview by October 11th. An e-mail with information about video presentations for the virtual workshop will follow soon.

Please contact us if you have any questions.